# The Threshold Effect of FDI on CO_2_ Emission in Belt and Road Countries

**DOI:** 10.3390/ijerph19063523

**Published:** 2022-03-16

**Authors:** Ying Nie, Qingjie Liu, Rong Liu, Dexiao Ren, Yao Zhong, Feng Yu

**Affiliations:** 1Institute of Data Science and Agricultural Economics, Beijing Academy of Agriculture and Forestry Sciences, Beijing 100097, China; niey@agri.ac.cn (Y.N.); zhongy@agri.ac.cn (Y.Z.); yuf@agri.ac.cn (F.Y.); 2Belt and Road School, Beijing Normal University, Beijing 100875, China; 3Institute of Geographical Sciences and Natural Resources Research, Chinese Academy of Sciences, Beijing 100101, China; 4School of Management, Guangzhou College of Technology and Business, Guangzhou 510850, China; dx0703@163.com

**Keywords:** FDI, CO_2_ emissions, Belt and Road countries, PSTR model, threshold effect

## Abstract

Under the background of the global “carbon neutrality” goal, it is of great significance to study the environmental effect of FDI in rapid economic development. This paper proposes an original framework to determine the relative influence of five factors on the Belt and Road countries with a strong FDI-CO_2_ association. Based on the panel smooth transition regression (PSTR) model, we establish country-specific and time-specific FDI-CO_2_ coefficients for 59 Belt and Road countries during 2003–2018. These coefficients are assumed to change smoothly as a function of five threshold variables, considered the most important in the literature devoted to the FDI-CO_2_ correlations. The results show that the degree of GDP per capita, industrialization, openness, and total factor productivity significantly influences the FDI-CO_2_ relationship. However, they showed obvious heterogeneity. The coefficient of elasticity of the environmental effects of FDI smoothly transitions between the different intervals, the relationship between GDP per capita and FDI-CO_2_ coefficient shows a bell-shaped change, the relationship between degree of trade openness and FDI-CO_2_ coefficient also shows a bell-shaped change, the relationship between industrialization level and FDI-CO_2_ coefficient shows an inverted N-shaped change, the change of a country’s technological level shows a bell-shaped relationship with the FDI-CO_2_ coefficient. The results indicate that PSTR model can be used to study the threshold effect on FDI’s influence on carbon dioxide emissions and the individual and time differences in coefficients of elasticity, to provide a new research perspective and new conclusions on the environmental effect of FDI in rapid economic development.

## 1. Introduction

Since China announced the Belt and Road Initiative (BRI), countries along the Belt and Road (hereafter “Belt and Road countries”) have made great efforts to attract Foreign Direct Investment (FDI) for economic growth. In 2018, the Belt and Road countries attracted a total of USD 7.13 trillion of FDI, accounting for 22.1% of total global FDI, compared to USD 5.3 trillion in 2013, an increase of 34.7%. Most Belt and Road countries are middle-income countries, which means that these countries are faced with dilemmas of insufficient capital resources, and FDI inflows provide these countries with the funds needed for development. At the same time, positive externalities in technology and management experience that can increase labor productivity and reduce costs [1].

However, as the FDI in Belt and Road countries has increased, their environmental issues have been exacerbated. Meanwhile, environment issues continue be the focus among the Belt and Road countries. The World Bank’s latest data show that the total amount of carbon emissions in the Belt and Road countries were 20.42 billion tons in 2018, accounting for 59.99% of world’s total carbon emissions, with China and India’s carbon emissions accounting for almost 40% of the global total amount. The carbon emissions of countries along the Belt and Road increased 15.08 times from 1.27 billion in 1990 to 20.42 billion in 2018. So, what role does FDI play in carbon emissions in the Belt and Road countries? This is the question this paper attempts to answer.

Whether FDI inflows is responsible for the increase in carbon dioxide emissions in a country has not been agreed upon. Azam, M. and Raza, A. (2021) empirically explore the interrelationship between foreign capital flows and environmental quality measured by trade-adjusted consumption-based carbon dioxide emissions for a panel of 125 countries in 1990–2018. The results obtained using system GMM analysis show that FDI has a significantly positive link with carbon dioxide emissions in Asia and Africa, but the links between these two variables are insignificant in the Latin American, Caribbean, and European regions. In the cases of the full-sample and developing countries, a significantly positive relationship is found between FDI and carbon dioxide emissions [2]. Some scholars believe that the existing research ignores the indirect impact of FDI on pollution, when examining pollution heaven and halo hypothesis, there are different conclusions [3]. Generally, there are two views: pessimism and optimism. Optimists believe that FDI inflows inhibit carbon emissions [4], while pessimists believe that FDI inflows increase carbon emissions [5]. However, the consensus is that FDI has a significant impact on carbon emissions [6,7]. So, this paper studies the relationship between FDI inflows and carbon emissions in the Belt and Road countries, which helps these countries to formulate more reasonable policies to guide FDI in favor of emission reduction.

## 2. Literature Review and Theoretical Framework

Existing literature can be divided into two main perspectives on the relationship between FDI and carbon dioxide emissions. The first concerns the impact of FDI on environmental pollution and adheres to the pollution haven hypothesis, which argues that FDI inflows lead to the deterioration of the host country’s environment. Walter (1979) and Pethig (1976) were the first to propose this hypothesis [8,9]. They stated that when countries relax their environmental regulations to develop their economies, and they encourage high-polluting and high-energy consuming industries to relocate and invest in those countries, resulting in substantially higher pollution emissions. Many scholars subsequently empirically tested this hypothesis, and results show that FDI inflows increases the emission of environmental pollutants of host countries. Liu Y. et al. (2017) investigated foreign investment on environmental quality in China using a carefully designed framework of a two-equation model for the period between 2002 and 2015, supported the pollution haven [10]. Nasir, M. A. et al. (2019) employed a set of quantitative techniques for panel data analysis to analyze the relationship between FDI and carbon dioxide drawing on the data from 1982 to 2014 in the selected ASEAN-5 economies. Their findings indicate that FDI leads to an increase in environmental degradation [11]. Shahbaz, M. et al. (2019) decomposes the environmental Kuznets curve into the scale, technique, and composition effects. They find out increases in FDI hamper environmental quality by increasing carbon emissions through empirical evidence [4].

The other view on the impact of FDI on environmental pollution is more optimistic. It holds that FDI inflow effectively reduces host country’s pollutant emissions, known as the pollution halo hypothesis. Birdsall and Wheeler (1993) were the first to propose this view. They stated that FDI brings high standards mode of production and advanced technology to host countries, which helps to reduce pollutant emissions [5]. Pao and Tsai (2011) examined the impact of FDI on carbon emissions in emerging market countries and found that FDI has reduced carbon emissions significantly [12]. Zhu et al. (2016) also found the inhibitory effect of FDI on host countries’ emissions of environmental pollutants with quantile regression method [13]. Zhang and Zhou (2016) obtained the empirical evidence supporting the pollution halo hypothesis with carbon emissions as the proxy variable of pollutant [14]. Liu et al. (2017) and Sung et al. (2018) found that FDI inflows reduced carbon emissions significantly with case of China through the spatial panel model [10,15]. Xu et al. (2019) focused on air pollutants and verified that FDI not only promotes applications of environmentally friendly technology but also effectively strengthens local environmental protection and supervision [16].

The above research on the impact of FDI on carbon dioxide emissions tended to focus on the linear relationship between the two factors [17]. However, the focus on recent research has shifted to the non-linearity of FDI and carbon dioxide emissions. The baseline idea is very simple: common knowledge that FDI inflow depends on other exogenous variables (trade openness, country size, etc.), which clearly matches the definition of a threshold regression model—threshold regression models specify that individual observations can be divided into classes based on the value of an observable variable [18]. Pasquale Pazienza (2019) introduced the squared terms of FDI to test whether FDI has a U-shaped nonlinear relationship with carbon emissions [6]. Sarkodie and Strezov (2019) constructed a third-order polynomial model to test the nonlinear effects of FDI on carbon emissions considering terms of square and cube of FDI in the model [19]. Xie et al. (2019) considered the nonlinear relationship between FDI and carbon emissions using the PSTR model and examined the direct and spillover effects of FDI inflows on carbon emissions at different threshold intervals [20].

In addition, previous research shows that the impact of FDI on pollutant emissions may be different with different stages of economic development, degree of trade openness, technology, and industrial level of host countries [21].

First, the impact of FDI on carbon emissions is affected by the stage of economic development of host countries. Aneta Kosztowniak (2016) studied the relationship between FDI and GDP in Poland from 1992 to 2012 by employing the Vector Error Correction Method impulse responses and variance decomposition analysis. They ultimately confirmed the bi-directional relationships between FDI and GDP in Poland [22]. Whether the impact of GDP on attracting FDI is stronger than that of FDI on GDP depends on the national conditions of different countries [23]. Developing countries attached great importance to the introduction of FDI in order to promote their technological upgrading and economic development [24]. These countries attract high-polluting FDI due to their imperfect environmental pollution supervision system and lax legal supervision, which serves to aggravate their environmental pollution problem [25]. With GDP rising in more emerging economies and their income levels approaching that of developed countries, the effect of relying on attracting FDI to promote economic growth is not very obvious, and the attractiveness for FDI decreases [26]. In comparison, when introducing investment, emerging economies generally choose clean FDI [27]. This indicates that the impact of FDI on CO_2_ emissions may have a threshold effect. When the economy and society continue to develop, the relationship between FDI and CO_2_ emissions also changes.

Second, host countries with different degree of trade openness have different effects of FDI inflows on carbon emissions. Carbon dioxide emissions and FDI have a cointegration relationship with trade having a one-period lag. These variables are different in different situations and countries [28]. Managi et al. (2009) believe that the impact of trade openness on carbon dioxide emissions depends on the pollutants and country choices [29]. Some studies decomposed the environmental effects of trade openness into scale effects, structural effects, and technological effects. Based on these different effects, some scholars believe that different effects play a major role in different periods. In the initial stage of trade openness, scale economy plays a role in attracting FDI, resulting in increased carbon dioxide emissions [30]. With the continuous development of foreign trade, structural effects and technical effects play a major role in attracting clean FDI and reducing carbon dioxide emissions [31]. Therefore, the relationship between FDI and carbon dioxide is also nonlinear with different degree of trade openness of host countries.

Third, the improvement of total factor productivity also plays a certain role in promoting carbon emission reduction by attracting FDI with high level of technology. Pan et al. (2018) argued that FDI has provided both advanced management experience and production technology for economies [32]. The inflow of FDI is also playing an increasingly important role in technological progress. FDI improved the technology factor productivity by introducing advanced technologies [33]. Advanced technology emits less CO_2_ emissions [34], nonlinear relationship between technology factor productivity and CO_2_ emission [35], but there is still a nonlinear relationship between FDI and technology factor productivity [36]. Moreover, industrial total factor productivity is often used as a comprehensive indicator to measure the technological level, and the interaction between industrial total factor productivity and the spillover effect of FDI has been controversial. Hui, W. et al. (2020) examined the main effect between two-way FDI and LCTFP on the basis of the data of 33 industries from 2004 to 2017 in China. They find the relationship between FDI and LCTFP was heterogeneous under pollution-intensive industries and relatively clean industries, as well as high and low environmental policy uncertainty [37]. Zhang, S. et al. (2021) analyzes the impact of FDI quantity and quality on the low-carbon development of the STPs based on the data of 52 STPs in China from 2011 to 2018, using Hansen’s nonlinear panel threshold regression model. The results show There is a nonlinear relationship between FDI and total factor productivity [38].

Fourth, the Belt and Road Countries have entered the age of industrialization, the continuous growth of FDI has greatly promoted the industrialization level of these countries. Host countries at different stages of industrialization attract different types of FDI. This has an all-around impact on the country’s economic and industrial development. It can not only bring changes in the GDP growth rate and internal growth source but also change the quantity and composition of carbon emissions in the development process [39]. The conditional roles that a country’s industrialization levels may play in the impact the FDI on carbon emissions [40]. Bai et al. (2020) investigated the effects of FDI on carbon productivity and the industrialization on FDI carbon productivity in 30 provincial-level regions of China from 2003 to 2017 by employing the traditional panel regression model, a panel quantile model and panel threshold model Their empirical result identified that the relationship between FDI and carbon productivity is affected by the level of industrialization, and there is a significant nonlinear threshold effect during the sample period [41].

In this paper, we investigate the potential threshold effects in the relationship between FDI and CO_2_ emission. Thus, we propose to test the relevance of FDI regression parameters (or FDI-CO_2_ coefficients) into classes given the values of four main factors generally quoted in this literature by Panel Smooth Transition Regress (PSTR) model: (i) GDP per capita, (ii) degree of trade openness, (iii) industrialization, and (iv) total factor productivity. Based on PSTR specifications, the paper derives FDI-CO_2_ coefficients, which vary not only between countries but also with time. Thus, it provides a simple parametric approach to capture both cross-country heterogeneity and time variability of the FDI-CO_2_ correlations. Additionally, the approach allows for smooth changes in country-specific correlations depending on a threshold variable. Consequently, we consider the four potential threshold variables previously mentioned as potential explanations of the cross-country heterogeneity and/or the time variability of FDI-CO_2_ coefficients for the Belt and Road countries.

The rest of this paper is organized as follows: In the next section, we discuss the threshold specification of the regression model and, in particular, the cross-country heterogeneity and the time variability of FDI coefficients. The choice of the threshold variables, linearity tests, and estimates for the parameters are then presented in a third section. The fourth part of the paper is data description. The fifth part is given over to the results of the linearity tests and the estimates obtained from various panel threshold models. Finally, based on these PSTR estimates, we calculate the individual FDI parameters and discuss the relative influence of the various threshold variables. The last section concludes.

## 3. FDI-CO_2_ Relationship: Toward a Threshold Specification

The basis of our empirical approach consists of evaluating CO_2_ emissions for a panel of N countries. The corresponding model is then defined as follows:(1)$$ln\mathrm{CO}2_{\mathrm{it}}=\alpha_{i}+\beta lnFDI_{\mathrm{it}}+\varepsilon_{\mathrm{it}}\;.$$
where $ln\mathrm{CO}2_{\mathrm{it}}$ is the natural logarithm of carbon emissions observed for the *i*th country at time *t*, $lnFDI_{\mathrm{it}}$ is the natural logarithm of the net inflow of foreign direct investment, and $\alpha_{i}$ denotes an individual fixed effect. The residual $\varepsilon_{\mathrm{it}}$ is assumed to be *i.i.d.* (0,$\sigma_{\varepsilon }^{2}$). According to the existing research, there may be a nonlinear relationship between CO_2_ and FDI. Equation (1) sets the impact coefficient of *FDI* as β, and considers that the impact of FDI on CO_2_ emissions in any country at any time is the same, which is not realistic.

To solve the problem of the heterogeneity of influence coefficients, researchers usually use group regression to estimate the regression coefficients of different samples. The advantage of this method is that it is simple and easy to generalize. The disadvantages are that it is difficult to determine standards for grouping, and it is easy to lose some common information between samples when creating several sub-samples for regression analysis. In addition, sub-sample regression separates the transition process between samples, which does not conform with reality. To overcome the first shortcoming of group regression, Bruce and Hansen (1999) proposed Panel Threshold Regression (PTR) [21]. In this method, the transition mechanism between extreme regimes is very simple: at each date, if the threshold variable observed for a given country is smaller than a given value, called the threshold parameter, CO_2_ emission is defined by a particular model (or regime). For instance, let us consider a PTR model with two extreme regimes, as illustrated by Equation (2): (2)$$ln\mathrm{CO}2_{\mathrm{it}}=\alpha_{i}+\beta_{0}lnFDI_{\mathrm{it}}+\beta_{1}lnFDI_{\mathrm{it}}g\left(q_{\mathrm{it}};c\right)+\varepsilon_{\mathrm{it}}\;,$$
where $q_{\mathrm{it}}$ denotes a threshold variable, c is a threshold parameter and where the transition function $g\left(q_{\mathrm{it}};c\right)$ corresponds to the indicator function with Equation (3) as follows:(3)$$g\left(q_{\mathrm{it}};c\right)=\left\{\begin{array}{c} 1\;\;\mathrm{if}\;q_{\mathrm{it}}\geq c\;\\ 0\;\mathrm{otherwise} \end{array}\right..$$

With such a model, the FH coefficient is equal to $\beta_{0}$ if the threshold variable is smaller than c and is equal to β_0_ + β_1,_ if the threshold variable is larger than *c*. This model can be extended to a more general specification with r regimes. However, even in this case, the PTR model imposes the constraint that the value of the FDI-CO_2_ coefficient can be divided into a (small) finite number of classes. Such an assumption may be unrealistic even for the Belt and Road countries.

To solve this problem, Gonzalez et al. (2005) extended the PTR model to create a panel smooth transition regression (PSTR) model. This model introduced a smooth transition function to achieve the transition between different sample categories [42]. As a result, the PSTR model has advantages in terms of grouping samples and achieving smooth transitions between groups [43]. This article uses the PSTR model to study the impact of FDI on carbon dioxide emissions in Belt and Road countries under different threshold variables. Based on the above analysis, the following PSTR model can be created in Equation (4):(4)$$ln\mathrm{CO}2_{\mathrm{it}}=\alpha_{i}+\beta_{0}ln{\mathrm{FDI}}_{\mathrm{it}}+\sum_{j=1}^{r}\beta_{j}ln{\mathrm{FDI}}_{\mathrm{it}}g_{j}\left(q_{\mathrm{it}};\gamma_{j};c_{j}\right)+\varepsilon_{\mathrm{it}},$$
wherein, $\beta_{0}$ and $\beta_{j}$ are the effects of linear and non-linear parts, respectively; $\mu_{i}$ is the individual fixed effect; $\varepsilon_{\mathrm{it}}$ is the random error; $g_{j}\left(q_{\mathrm{it}};\gamma_{j};c_{j}\right)$ is the transition function, with the number determined by the parameters. The specific form of the transition function is as follows:

The r transition functions $g_{j}\left(q_{\mathrm{it}};\gamma_{j};c_{j}\right)$ depend on the slope parameters $\gamma_{j}$ and on location parameters $c_{j}$. In this generalization, if the threshold variable $q_{\mathrm{it}}$ different from $ln{\mathrm{FDI}}_{\mathrm{it}}$, the FDI-CO_2_ coefficient for the i*_th_* country at time t is defined by the weighted average of the r + 1 parameters $\beta_{j}$ associated with the r + 1 extreme regimes:(5)$$e_{\mathrm{it}}=\frac{\partial ln\mathrm{CO}2_{\mathrm{it}}}{\partial ln{\mathrm{FDI}}_{\mathrm{it}}}=\beta_{0}+\sum_{j=1}^{r}\beta_{j}g_{j}\left(q_{\mathrm{it}};\gamma_{j};c_{j}\right).$$

The coefficient of each period and each individual is a continuous function of the conversion variable. Through the analysis of the changing relationship between $\beta_{\mathrm{it}}$ and $q_{\mathrm{it}}$, it is possible to test whether the different degree of economic development, openness, industrialization, and technology factor productivity of the host country has a significant impact on the environmental effects of FDI. As reflected in the transition function of Equation (5), g is a function of q. When q changes, g smoothly changes to between 0 and 1. In other words, the coefficient $\beta_{\mathrm{it}}$ of $ln{\mathrm{FDI}}_{\mathrm{it}}$ changes between $\beta_{0}$ and $\left(\beta_{0}+\sum_{j=1}^{r}\beta_{j}\right)$, which is equivalent to the weighted average of the sum of $\beta_{0}$ and $\beta_{j}$. Taking r = 1 as an example, when $\beta_{j}>0$,then $\beta_{0}<\beta_{\mathrm{it}}<\beta_{0}+\beta_{j}$. This shows that the influence coefficient of FDI on carbon dioxide emissions in Belt and Road countries increases along with increases in the country’s economic development level (or openness, industrialization, population density, and technology factor productivity). When $\beta_{j}<0$, then $\beta_{0}+\beta_{j}<\beta_{\mathrm{it}}<\beta_{0}$, which shows that the influence coefficient of FDI on carbon emissions in Belt and Road countries decreases as the economic development level (or openness, industrialization, and technology factor productivity) of the countries increases. As a result, $\beta_{0}$ shows the initial effect of FDI on carbon emissions in the host country, and $\beta_{j}$ shows the impact of FDI on carbon emissions in the host country regarding its threshold variables and its time-varying non-linear characteristics.

So, based on the PSTR Model (see Equation (4)), we consider four threshold variables, thus forming four groups of models. 

Model A is as follows: (6)$$ln\mathrm{CO}2_{\mathrm{it}}=\alpha_{i}+\beta_{0}ln{\mathrm{FDI}}_{\mathrm{it}}+\sum_{j=1}^{r}\beta_{j}ln{\mathrm{FDI}}_{\mathrm{it}}g_{j}\left(lnPGDP_{\mathrm{it}};\gamma_{j};c_{j}\right)+\varepsilon_{\mathrm{it}}.$$

Model B is as follows:
(7)$$ln\mathrm{CO}2_{\mathrm{it}}=\alpha_{i}+\beta_{0}ln{\mathrm{FDI}}_{\mathrm{it}}+\sum_{j=1}^{r}\beta_{j}ln{\mathrm{FDI}}_{\mathrm{it}}g_{j}\left(lnOPEN_{\mathrm{it}};\gamma_{j};c_{j}\right)+\varepsilon_{\mathrm{it}}.$$

Model C is as follows:(8)$$ln\mathrm{CO}2_{\mathrm{it}}=\alpha_{i}+\beta_{0}ln{\mathrm{FDI}}_{\mathrm{it}}+\sum_{j=1}^{r}\beta_{j}ln{\mathrm{FDI}}_{\mathrm{it}}g_{j}\left(lnIND_{\mathrm{it}};\gamma_{j};c_{j}\right)+\varepsilon_{\mathrm{it}}.$$

Model D is as follows:(9)$$ln\mathrm{CO}2_{\mathrm{it}}=\alpha_{i}+\beta_{0}ln{\mathrm{FDI}}_{\mathrm{it}}+\sum_{j=1}^{r}\beta_{j}ln{\mathrm{FDI}}_{\mathrm{it}}g_{j}\left(lnTFP_{\mathrm{it}};\gamma_{j};c_{j}\right)+\varepsilon_{\mathrm{it}}.$$

In the first model (called Model A, see Equation (6)), we assume that the conversion mechanism in the carbon emission equation is determined by GDP per capita. We expect that the higher the GDP per capita level the lower the coefficient of the impact of FDI on carbon emissions will be. This may be because more attention is paid to the environmental impact in the process of attracting FDI with the continuous improvement of a country’s economic development level. Therefore, the stimulating effect of the increased FDI on carbon emissions in countries with relatively more developed economies is reduced. In the second equation (Model B, see Equation (7)), the impact coefficient of FDI on carbon emissions is affected by the degree of trade openness of host countries. We expect that the higher the level of openness, the more likely it will attract FDI that is not conducive to environmental development, thus enhancing the stimulating effect of FDI on carbon emissions. However, as the level of openness increases to a certain stage, the promoting effect gradually decreases. The third equation (Model C, see Equation (8)) shows that the impact of FDI on carbon emissions is affected by the industrialization level of host countries. The higher the level of industrialization is, the weaker the positive stimulus of FDI on carbon emissions is. The fourth equation (Model D, see Equation (9)) targets how the impact of FDI on carbon emissions is affected by a country’s technical level: the higher the country’s technical level, the weaker the stimulating effect of FDI on carbon emissions may be.

## 4. Data Description

This study concerns a selection of 59 Belt and Road countries during 2003–2018. The 59 Belt and Road countries were selected in Central and Eastern Europe, Southeast Asia, West Asia, North Africa, South Asia, and Central Asia (shown in Table 1). Our data are taken from the Penn World Tables, World Development Indicator and UNCTAD (see Table 2).

## 5. Discussion

### 5.1. Estimation and Specification Tests

Before conducting the PSTR model estimation, a linear test must be performed to determine the potential existence of a non-linear relationship between the variables. The null hypothesis of the linear test is that Model (3) is a linear model, and the alternative hypothesis is that Model (3) contains at least one smooth nonlinear transition function. That is, if r = 0, then the variables in the model do not have non-linearity, and if r ≥ 1, then the model is nonlinear, and the PSTR model should be used for regression. The linear test is mainly used to analyze whether the elastic constants are homogeneous, to determine whether to choose a linear model. In the model, to test whether the linear hypothesis is true or not, we follow the approach of Luukkonen et al. (1988) and use the first-order Taylor series expansion of the transition function near γ = 0 to replace the transition function in the model, which yields the following regression model (i.e., Equation (10)) [45]:(10)$$ln\mathrm{CO}2_{\mathrm{it}}=\alpha_{i}+\beta_{0}ln{\mathrm{FDI}}_{\mathrm{it}}+\beta_{1}ln{\mathrm{FDI}}_{\mathrm{it}}q_{\mathrm{it}}+\varepsilon_{\mathrm{it}}.$$

The linear test is equivalent to testing H0: $\beta_{1}=0$. Following the method of Colletaz and Hurlin (2008), the construction statistics are as follows [46]:(11)$${\;\mathrm{LM}}_{F}=\left({\mathrm{SSR}}_{0}-{\mathrm{SSR}}_{1}\right)/\left[{\mathrm{SSR}}_{0}/\left(\mathrm{TN}-N-1\right)\right].$$

Under the null hypothesis, the F-statistic has an approximate F (1, TN−N−1) distribution. In Equation (11), ${\mathrm{SSR}}_{0}$ and ${\mathrm{SSR}}_{1}$ are the residual sum of squares of the linear panel model and the PSTR model when r = 1; and N is the number of individuals in the panel dataset. The original hypothesis of the linear test is that the linear model should be selected, and the alternative hypothesis is that the PSTR model is most appropriate. If the test result considers the PSTR model to be appropriate, then a remaining non-linearity test must be performed to determine the number of groups. The test starts with the inspection of H0: r = 1, Ha: r = 2. If H0: r = 1 is rejected, the next step is to inspect H0: r = 2, Ha: r = 3. This procedure is completed until H0: r = r* is acceptable and it is determined that the model has r* + 1 categories. After determining the number of categories, a model is constructed, and a nonlinear least squares (NLS) estimation is conducted. This study uses MATLAB2020a to test the PSTR model. The test results are shown in Table 3.

The results of these linearity tests and specification tests with no remaining non-linearity are reported in Table 3. For each specification, we calculate the statistics for the linearity tests LM_F_ (H_0_: r = 0 versus H_1_: r = 1) and for the tests of no remaining non-linearity LM_F_ (H_0_: r = a versus H_1_: r = a + 1). Firstly, the four models reject the original assumption, and the nonlinear model is more suitable. Further tests show that for Model A and B, the optimal (LM_F_ criterion) number of threshold functions is r = 1, and for Model C and D, the optimal number is r = 2. The level of economic development, industrialization, trade openness, and technological level may have an impact on the FDI-CO_2_ coefficient, which shows heterogeneity. In terms of the optimal number of locational parameters, select m = 2 if the rejection of $H_{0}^{2}$ is the strongest one, otherwise select m = 1. In Model A, m = 2 is chosen. In the other three models, the null hypothesis is models, so m = 1 is chosen. From the above PSTR model test results, the FDI-CO_2_ coefficient may show heterogeneity characteristics because of a country’ s economic development level, degree of trade openness, industrialization level and technical level.

### 5.2. PTSR Model Results and Discussion

Based on the above linearity test and remaining non-linearity test results, PSTR model with a transition function and location parameter can be constructed. The optimal selection of Model A and B is r = 1, m = 2, the optimal selection of Model C and D is r = 2, m = 1. The PSTR model is used for regression, with the threshold variable to estimate the nonlinear relationship between FDI and carbon emissions, if this relationship smoothly transitions between different groups. First, a linearity test and remaining non-linearity test are performed, and nonlinear least square (NLS) is used to estimate the parameters of the PSTR model. Table 4 shows the estimation results of the PSTR model.

Table 4 shows the estimated parameters of the final PSTR model. First, we can see that the estimated slope parameter γ in all models is relatively small, which means the transfer function $g\left(q_{\mathrm{it}};\gamma ;c\right)$ cannot be simply expressed as a threshold model with an index function. The transformation between the thresholds of the model is smooth rather than sudden change at a certain breakpoint. The four models move from the low threshold interval to the high threshold interval at a relatively smooth and slow speed, and the transformation speed is moderate. That is to say that the impact of FDI on carbon emissions cannot be simply classified into a few categories. The FDI-CO_2_ coefficient in 59 BRI countries for 2003–2018 is a series of continuous values. This also shows that it is inappropriate to simply analyze the impact of FDI on carbon emissions by a linear relationship. Secondly, and most importantly, we can assess the impact of four threshold variables on the FDI-CO_2_ coefficient. Figure 1 shows the relative importance of the influence of different threshold variables on the FDI-CO_2_ coefficient. It can be seen from these figures that the four groups of FDI-CO_2_ coefficients from the PSTR model may take different continuous values, and the coefficient values are also changing with the change of threshold variables.

The FDI-CO_2_ coefficient varies from different threshold ranges. In Model A, GDP per capita is the threshold variable. The estimated FDI-CO_2_ coefficient changes show a bell-shaped curve, and the impact coefficient fluctuates between 0.18 and 0.25. In the early stage of economic development, economic growth stimulated FDI to promote carbon emissions. This may be because less developed countries do not consider whether FDI is clean for economic growth, but blindly pursue large amounts of investment to drive economic development. This will lead to the result that a large number of unclean FDI flows into the host country. The faster the economic growth is, the faster the carbon emission growth caused by FDI is. When country enter a more developed stage, the country’s strategy of attracting foreign investment has been adjusted, and began to consider the balance between environmental protection and economic growth. At this period, the quality of FDI attracted is relatively high, and the growth of carbon emissions caused by it began to slow down. In Model B, the relationship between the degree of trade openness and FDI-CO_2_ coefficient also shows a bell-shaped curve. In the period of lower degree of trade openness, the attraction of the country to FDI is limited, and it is easy to become a pollution paradise, that is, to attract unclean FDI, so that the stimulating effect of FDI on carbon emissions is enhanced. With the continuous improvement of the trade openness of host countries, and the attraction strategy of FDI is adjusted, which is beneficial to the development of the environment. In this case, clean FDI is attracted, thus effectively reducing its stimulating effect on carbon emissions.

In Model C, the relationship between the level of industrialization and the coefficient of FDI-CO_2_ shows an inverted N-shaped type change, that is, in the early stage of industrialization, with the increase in the level of industrialization, the stimulating effect of FDI on carbon emissions being weaker. When industrialization develops to a certain stage, the stimulating effect of attracting FDI on carbon emissions begins to increase rapidly. When the development of industrialization continued, this stimulating effect is alleviated. This reflects the heterogeneity of environmental pollution effect caused by different types of FDI attracted by countries at different stages of industrialization. With the continuous development of industrialization, the industrial technology level of foreign investment is improved, energy consumption is gradually reduced, and the promotion effect on carbon emissions is also reduced. In Model D, in the process of continuous improvement of a country’s total factor productivity, the promoting effect of FDI on carbon emissions first increases and then decreases. Only when the national technology reaches a higher stage, the technical level begins to pay more attention to the improvement of energy consumption and carbon emission technology, thus decreasing the FDI-CO_2_ coefficient.

### 5.3. Nonlinear Marginal Analysis of the PSTR Model

To further analyze the individual and temporal effects of FDI inflows on carbon dioxide emissions, this study conducts a non-linear marginal analysis of the PSTR model to calculate the FDI-CO_2_ coefficient. The advantage of the PSTR model is that it can detect individual characteristics and dynamic changes over time in the dataset. According to Equation (3), the average value of FDI-CO_2_ coefficients of 59 countries along the Belt and Road from 2003 to 2018 under four sets of threshold variables can be obtained.

First, as shown in Figure 2, a country at different stages of economic development will also significantly affect the FDI-CO_2_ coefficient. The study finds that the change of GDP per capita and the FDI-CO_2_ coefficient shows an obvious bell-shaped relationship. When natural logarithm of GDP per capita is lower than 8.2, the increased of economic development level promotes the carbon emission effect of FDI. When natural logarithm GDP per capita is higher than 8.2, continuing economic development is conducive to alleviating the carbon emission effect of FDI. For example, the natural logarithm of GDP per capita of Mongolia is 7.749, which is much higher than that of Nepal (6.370). The impact coefficient of FDI on carbon emissions in Mongolia is 0.239, which is also higher than that of Nepal (0.213). Attracting a unit of FDI will increase Mongolia’s carbon emissions by 0.239 units and Nepal’s carbon emissions by 0.213 units. Countries at this stage also include Laos, India, Kyrgyzstan, Tajikistan, and Bangladesh. Their economic development is not mature enough, and their further economic growth is at the expense of the environment. With the continuous maturity of economic development, the country pays more attention to the quality of FDI. Considering the environmental sustainability, FDI with high environmental technology is attracted, so the stimulating effect of FDI on carbon emissions will be alleviated. For example, the GDP per capita of Slovenia is 10.001, which is higher than that of Belarus (8.524), Kazakhstan (8.911), and other countries, while the FDI-CO_2_ coefficient is the lowest in Slovenia, which is 0.192. Belarus and Kazakhstan are higher, which are 0.241 and 0.236, respectively. Slovenia’s carbon emissions increase 0.192 units, while Belarus and Kazakhstan’s carbon emissions increase 0.241 and 0.236 units, with a unit increased of FDI inflow.

In Figure 3, the higher the degree of trade openness, the higher the average value of FDI-CO_2_ coefficient when the natural logarithm of trade openness of host countries is below 4. For example, the average degree of natural logarithm of China’s openness from 2003 to 2018 is 3.835, the average value of the impact coefficient of FDI on carbon emissions is 0.244, and the average value of Pakistan’s openness is 3.397, which is lower than China’s openness level, and its FDI-CO_2_ coefficient is also lower than China, which is 0.234. Accordingly, China’s openness level is higher than Indonesia (3.730), Bangladesh (3.640), Egypt (3.545), and India (3.485), so the impact coefficient of FDI-CO_2_ is also higher than these countries. The elastic coefficient of FDI on carbon emissions in Indonesia is 0.242, Bangladesh is 0.240, Egypt is 0.238, and India is 0.236. When the degree of openness of a country is higher than four, this effect shows the opposite characteristics. The level of openness continues to increase, and the promotion of FDI on carbon emissions is weakened. For example, the level of openness of Croatia is 4.116, and the FDI-CO_2_ coefficient of the country is 0.249. In comparison, the level of openness of Bulgaria is higher than that of Croatia, which is 4.629, but its FDI-CO_2_ coefficient is 0.240, which is lower than that of Croatia. Therefore, as shown in Figure 3, the openness level of Jordan, Kyrgyzstan, Bulgaria, Thailand, Belarus, the Czech Republic, and Malaysia is gradually increasing, while the impact coefficient of the FDI-CO_2_ is gradually decreasing. In comparison, the higher the level of openness of these countries, the deeper their participation in the process of globalization, and the higher the quality of attracting FDI inflow. With the inflow of environmental-related high-tech FDI, its stimulating effect on carbon emissions is also decreasing. A country with this characteristic continues to choose an open development strategy, which is conducive to inhibiting the impact of FDI on carbon emissions and promoting environmentally sustainable development.

In Figure 4, the influence of industrialization levels on the FDI-CO_2_ coefficient shows an inverted N-shaped type change. From the average point of view, when the industrialization level (natural logarithm) is less than 3, the improvement of the industrialization level reduces the FDI-CO_2_ coefficient. Taking the Maldives as an example, its industrialization level is 2.332, which is lower than that of Nepal (2.679), but its influence coefficient of FDI on carbon emissions is 0.249, which is higher than that of Nepal (0.228). When the industrialization level is between 3 and 3.6, the improvement of industrialization level promotes the carbon emission effect of FDI. For example, Belarus’ s industrialization level is 3.554, and its influence coefficient of attracting FDI on carbon emissions is 0.258, much higher than that of Bulgaria (0.219), while Bulgaria’s industrialization level is 3.195, lower than that of Belarus. Similarly, the FDI-CO_2_ coefficient of Russia with a higher industrialization level is also higher than that of Singapore, Laos, Bangladesh, and other countries with lower industrialization levels. When the level of industrialization is higher than 3.6, its impact on the FDI-CO_2_ coefficient begins to decrease, which is negatively correlated. The higher the level of industrialization, the lower the promotion of FDI on carbon emissions, such as Kuwait with a higher level of industrialization, its FDI-CO_2_ coefficient is lower than Armenia, Turkmenistan, and China with a lower level of industrialization. 

The technical level represented by total factor productivity also has a significant impact on the carbon emission effect of FDI. Figure 5 shows that with the improvement of technical level, the FDI-CO_2_ coefficient increases first and then decreases. When the total factor productivity (natural logarithm) is lower than 6, the promotion effect of FDI on carbon emission is strengthened by the improvement of technical level. When the total factor productivity is higher than 6, the carbon emission effect of FDI is gradually weakened with the improvement of technical level. For example, countries such as India and Laos, which are on the left, have higher FDI-CO_2_ coefficients than countries such as Nepal and Tajikistan, which are relatively low in technology. Attracting a unit of FDI inflow will lead to an increase of 0.253 and 0.247 units in carbon emissions in India and Laos, while relatively lower level of technology will lead to an increase of 0.132 and 0.186 units in FDI-CO_2_ coefficients in Nepal and Tajikistan. In countries on the right of Figure 5, the coefficient of FDI-CO_2_ in Kuwait with higher technical level is lower than that in China with lower technical level. Similarly, attracting a unit of FDI will increase China’s carbon emissions by 0.256 units, while Kuwait will increase by 0.211 units. The reason for this phenomenon may be that the development of technology has periodic characteristics. In the early stage, the development of technology mainly focuses on production efficiency and ignores the impact on the environment. When technology develops to a certain extent, the state begins to pay attention to reducing environmental pollution while improving efficiency. Therefore, the improvement in technology in this period tends to save energy and reduce consumption, which alleviates the promoting effect of FDI on carbon emissions.

Another advantage of the PSTR model is that it can analyze time dynamics in the dataset. Thus, the average elasticity of FDI to carbon dioxide emissions of Belt and Road countries each year from 2003 to 2018 is calculated. Figure 6 is the average value of the FDI-CO_2_ coefficient of 59 Belt and Road countries corresponding to five threshold variables. Under the influence of trade openness, the FDI-CO_2_ coefficient is the highest, but the coefficient has been decreasing in recent years, followed by the influence of industrialization level and technology. The FDI-CO_2_ coefficient under the level of economic development is lowest.

Taking the FDI-CO_2_ coefficient estimated by GDP per capita as a threshold variable as an example, the influence coefficient of FDI on carbon emissions in 59 countries from 2003 to 2018 shown in Figure 7 is obtained to analyze the spatial and temporal variation characteristics.

Figure 7 shows that the impact coefficient of FDI on carbon emissions in 59 Belt and Road countries from 2003 to 2018 shows indigenous heterogeneity. From 2003 to 2018, the impact of FDI attracted by countries along the Belt and Road on carbon emissions shows three characteristics. The first is that the impact coefficient changes little, such as Albania, Armenia, Azerbaijan, Bosnia and Herzegovina, Bulgaria, Indonesia, China, Thailand, Sri Lanka, Macedonia, Maldives, and other countries. The promoting effect of FDI on carbon emissions in these countries has long been high, and the coefficient remains at a high level. The FDI-CO_2_ coefficient of the United Arab Emirates and Qatar continued to be low. The second is that the FDI-CO_2_ coefficient gradually decreases or increases, such as Bahrain, Estonia, Israel, Latvia, Saudi Arabia, Slovakia, and other countries, and the promoting effect of FDI on carbon emissions gradually decreases. In addition, Tajikistan, Vietnam, Uzbekistan, Laos, Kyrgyzstan, India, Bangladesh, Cambodia, Nepal, Pakistan, and other countries, FDI-CO_2_ coefficient gradually increased, attracting FDI is not conducive to mitigation of carbon emissions. The third is that the FDI-CO_2_ coefficient increases first and then decreases. For example, the FDI-CO_2_ coefficient in Yemen increases first and then decreases. In addition, Myanmar’s FDI-CO_2_ coefficient changed greatly around 2012, because Myanmar’s economic development level declined rapidly after 2012, from 2011 GDP per capita of USD 154,919 to USD 2209 in 2012. After that, the country sought economic recovery motivation to attract FDI, ignoring the control of carbon emissions, large inflows of unclean FDI lead to FDI-CO_2_ coefficient increased substantially.

## 6. Policy Implications

This study has important policy implications. Our research found that, in general, attracting FDI has not been conducive to improving environmental quality in Belt and Road countries, and it has led to an increase in carbon dioxide emissions, in line with the pollution haven hypothesis. The Belt and Road countries are generally minimally developed or developing countries, and they are largely unable to attract high-tech FDI inflow. As the economic development levels of Belt and Road countries increased, the promotional effect of FDI on carbon dioxide emissions has weakened. This may be because mature economies are better able to attract FDI and gradually attract more foreign investment in the field of renewable energy, thereby reducing the share of fossil fuels used in energy production and easing carbon dioxide emissions. Moreover, attracting FDI in the field of technology can lead to new technologies that completely change a country’s energy sector and reduce carbon dioxide emissions. As a result, it is necessary to continue to promote economic development, actively attract foreign investment in high-tech industries, encourage foreign companies to invest in renewable infrastructure and modern technology, urge governments to provide subsidies to guide multinational companies toward green and clean technologies, and guide FDI toward high-tech industries and high-efficiency production. Foreign investors must also be required to evaluate and publish information on carbon emissions during the investment process, strictly control carbon emissions, and promote environmental governance and sustainable development.

The limitation of this study is that in the process of exploring the heterogeneity of the FDI-CO_2_ coefficient, the overall scale of FDI is used, which is limited to data acquisition. This paper does not conduct in-depth research at the industry level. In fact, the impact of FDI on different industries on carbon emissions may be more realistic, and it will also make the mechanism of FDI-CO_2_ coefficient affected by a country’s economic level, industrialization level, degree of trade openness and technological level clearer. This is certainly our further research direction.

## 7. Conclusions

Based on a systematic review of existing literature, the researcher finds although there seems to be an agreement on the nonlinear impact of economic growth, FDI inflows, technology factor productivity, trade openness on environmental degradation. However, existing studies were limited because the evidence of the impact on economic growth, FDI inflows, technology factor productivity, industrialization, trade openness on environmental degradation appears to be mixed and inconclusive. There are two viewpoints between FDI and CO_2_: pessimism and optimism. Therefore, this paper takes economic growth as the threshold of the relationship between FDI and carbon dioxide emissions, and analyzes its threshold effect. Therefore, this paper takes economic growth, technology factor productivity, industrialization, trade openness as the threshold of the relationship between FDI and CO_2_, and analyzes their threshold effects. This paper used a research sample consisting of 59 countries situated along the Silk Road Economic Belt and the 21st-century Maritime Silk Road from 2003 to 2018 and constructed a PSTR model to study the nonlinear influence of FDI on carbon dioxide emissions. With the level of economic development, the degree of trade openness, the level of industrialization, and the technical level as the four threshold variables, the heterogeneity of the FDI-CO_2_ coefficient in different threshold intervals are investigated. It also used nonlinear marginal analysis to study the threshold effect on FDI’s influence on carbon dioxide emissions in Belt and Road countries and the individual and time differences in coefficients of elasticity, to provide a new research perspective and new conclusions on the influence of FDI on carbon dioxide emissions in Belt and Road countries.

The conclusions are as follows: First, FDI-CO_2_ coefficients are all positive, FDI has promoted carbon emissions in Belt and Road countries, in line with the pollution haven hypothesis. This may be because the majority of Belt and Road countries are developing countries, and the loosening of environmental regulations to attract FDI to develop their economies had a negative environmental effect. Second, the estimation results of the PSTR models show that the impact of FDI on carbon emissions in Belt and Road countries shows significant heterogeneity based on the degree of GDP per capita, industrialization level, trade openness, and total factor productivity of host countries. The coefficient of elasticity of the environmental effects of FDI smoothly transitions between the different intervals, In Model A, the relationship between GDP per capita and FDI-CO_2_ coefficient shows a bell-shaped change. When the level of economic development is in the underdeveloped stage, and the pursuit of economic development may be at the cost of losing the environment. When the economic development is gradually mature, the stimulating effect of attracting FDI on carbon emissions begins to ease, and the economy pursues clean and environmentally sustainable development. In Model B, the relationship between degree of trade openness and FDI-CO_2_ coefficient also shows a bell-shaped change. When the openness level is low, the country attracts FDI to pursue scale growth and neglects quality control, and the promotion effect of FDI on carbon emissions is increasing. With the deepening of globalization, the country begins to pay attention to environmental issues in the process of attracting FDI, thus alleviating the promotion effect of FDI on carbon emissions. In Model C, the relationship between industrialization level and the FDI-CO_2_ coefficient shows an inverted N-shaped change. Only when the development of industrialization into certain level, the industrial technology level of foreign investment is improved, the energy consumption is gradually reduced, and the promotion of carbon emissions is also reduced. In Model E, the change of a country’s technological level shows a bell-shaped relationship with the FDI-CO_2_ coefficient. In the early stages of a country’s technological level, the improvement of technology is not conducive to alleviating the promoting effect of FDI on carbon emissions. However, with the continuous maturity of technological level, advanced technologies with low energy consumption are introduced, and the energy consumption characteristics of FDI are diminished, so that the FDI-CO_2_ coefficient begins to decrease. The study findings provide a reference for accurately evaluating the potential pollution risks that may arise from attracting foreign investment and for effectively establishing the sustainable environmental policies in the Belt and Road countries during the postcrisis period.

## Figures and Tables

**Figure 1 ijerph-19-03523-f001:**
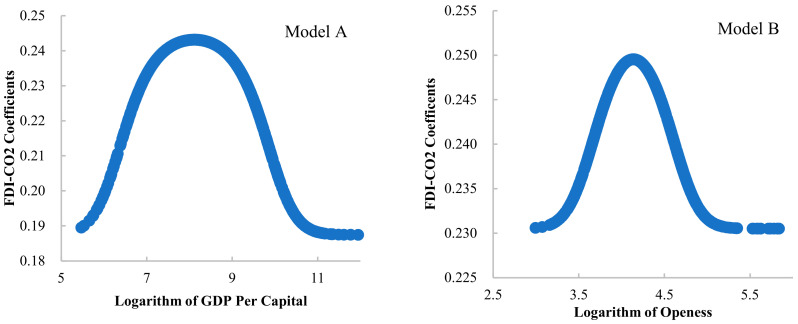
FDI-CO_2_ coefficients estimated by PSTR models with different threshold variables.

**Figure 2 ijerph-19-03523-f002:**
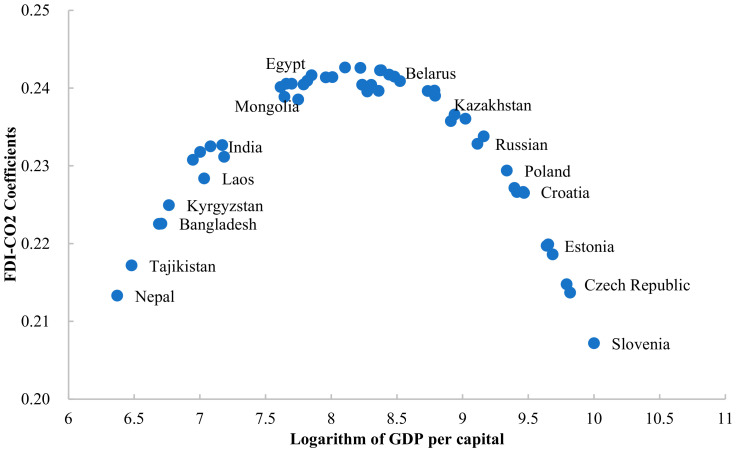
FDI-CO_2_ coefficients of Belt and Road countries (GDP per capita threshold).

**Figure 3 ijerph-19-03523-f003:**
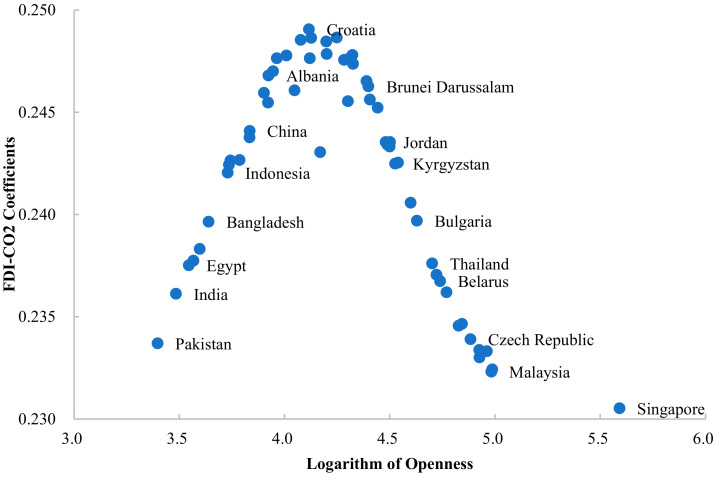
FDI-CO_2_ coefficients of Belt and Road countries (openness threshold).

**Figure 4 ijerph-19-03523-f004:**
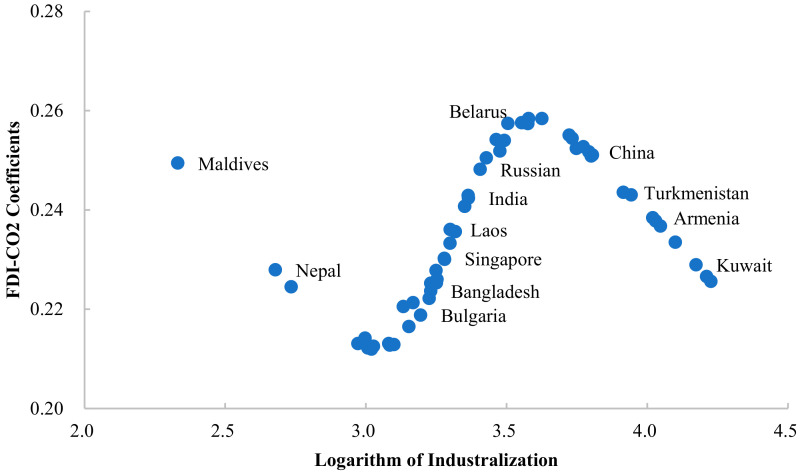
FDI-CO_2_ coefficients of Belt and Road countries (Industrialization threshold).

**Figure 5 ijerph-19-03523-f005:**
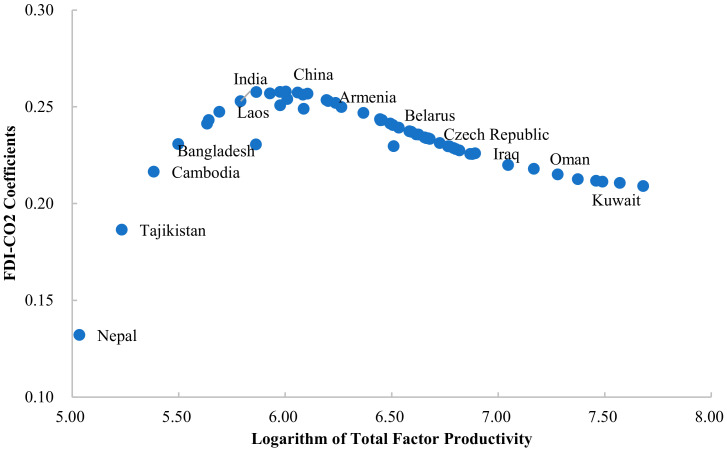
FDI-CO_2_ coefficients of Belt and Road countries (total factor productivity threshold).

**Figure 6 ijerph-19-03523-f006:**
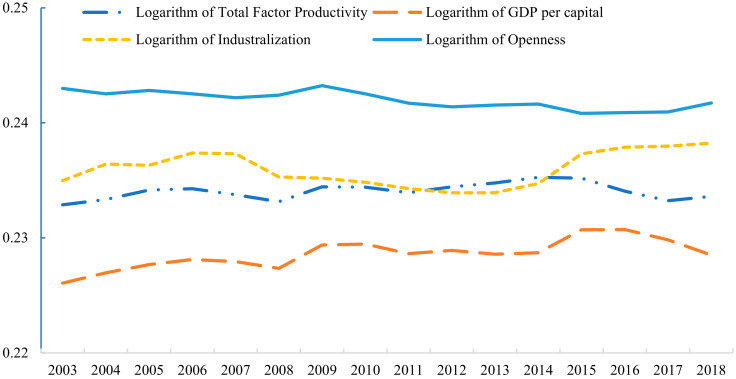
Temporal change in the coefficient of elasticity in Belt and Road countries.

**Figure 7 ijerph-19-03523-f007:**
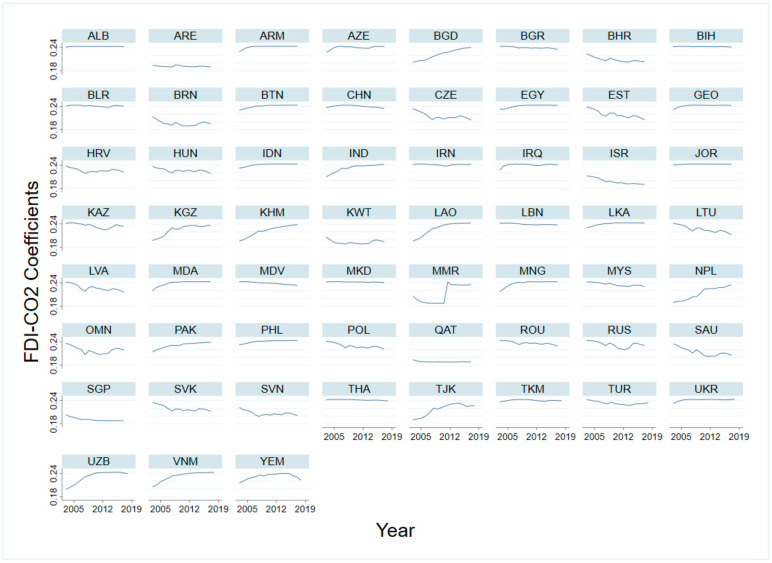
Estimated FDI-CO_2_ individual coefficients: PSTR Model A. Notes: All estimates are based on model A of Table 4. Abbreviations (listed in the order of the countries): ALB (Albania), ARM (Armenia), AZE (Azerbaijan), BGD (Bangladesh), BLR (Belarus), BTN (Bhutan), BIH (Bosnia), BRN (Brunei), BGR (Bulgaria), KHM (Cambodia), CHN (China), HRV (Croatia), CZE (Czech), EGY (Egypt), EST (Estonia), GEO (Georgia), HUN (Hungary), IND (India), IDN (Indonesia), IRN (Iran), IRQ (Iraq), ISR (Israel), JOR (Jordan), KAZ (Kazakhstan), KWT (Kuwait), KGZ (Kyrgyzstan), LAO (Laos), LVA (Latvia), LBN (Lebanon), LTU (Lithuania), MYS (Malaysia), MDV (Maldives), MNG (Mongolia), NPL (Nepal), MKD (North Macedonia), OMN (Oman), PAK (Pakistan), PHL (Philippines), POL (Poland), QAT (Qatar), MDA, (Republic of Moldova), ROU (Romania), RUS (Russia), SAU (Saudi Arabia), SGP (Singapore), SVK (Slovakia), SVN (Slovenia), LKA (Sri Lanka), TJK (Tajikistan), THA (Thailand), TUR (Turkey), TKM (Turkmenistan), UKR (Ukraine), ARE (United Arab Emirates), UZB (Uzbekistan), VNM (Vietnam), and YEM (Yemen).

**Table 1 ijerph-19-03523-t001:** 59 Belt and Road Countries.

Ch-Mon-Rus	Central Asia	West Asia and North Africa	Central and Eastern Europe	Southeast Asia	South Asia
China Mongolia Russian	Kazakhstan Kyrgyzstan Tajikistan Turkmenistan Uzbekistan	United Arab Emirates Oman Azerbaijan Egypt Pakistan Bahrain Georgia Qatar Kuwait Lebanon Saudi Arabia Turkey Armenia Yemen Iraq Iran Israel Jordan	Albania Estonia Belarus Bulgaria Bosnia Poland Czech Croatia Latvia Lithuania Romania North Macedonia Moldova Slovakia Slovenia Ukraine Hungary	Philippines Cambodia Laos Malaysia Myanmar Thailand Brunei Singapore Indonesia Vietnam	Bhutan Maldives Bangladesh Nepal Sri Lanka India

**Table 2 ijerph-19-03523-t002:** Property of the data.

Series	N	Obs.	Mean	Std.	Min.	Max.	Source
*lnCO2*	59	944	3.732	1.829	−1.171	9.210	World Development Indicator
*lnFDI*	59	944	9.606	2.344	−11.513	14.286	UNCTAD
*lnPGDP*	59	944	8.558	1.264	5.471	11.951	World Development Indicator
*lnIND*	59	944	3.428	0.402	2.087	4.315	World Development Indicator
*lnTFP*	59	944	6.433	0.616	4.853	7.933	Penn World Tables
*lnOPEN*	59	944	4.232	0.734	−1.624	5.839	World Development Indicator

Notes: Our data are issued from the Penn World Tables, World Bank World Development Indicator, and the UNCTAD statistics database. For the natural logarithm of carbon dioxide (lnCO2) emissions: the data are taken from the World Bank’s World Development Indicators (WDI) database. Carbon dioxide emissions are generated by the combustion of fossil fuels, including the consumption of solid, liquid, and gaseous fuels and combustion of natural gas, as well as during cement production. *lnFDI*: the data are derived from the latest information released by UNCTADstat database. The conversion variables (also the threshold variables) are: Level of economic development, which is represented by GDP per capita (*lnPGDP*) given in constant 2010 US dollars. The data come from the World BankWDI database; Industrialization level is the value added by industry as a percentage of GDP, the data derives from the WDI database; Trade openness (*lnOPEN*): exports plus imports divided by real gross domestic product per capital is the total trade as a percentage of GDP; Total factor productivity (*lnTFP*): According to the C-D production function including capital stock K, labor input L and output Y, TFP is inversely solved, that is $\mathrm{TFP}=Y/\left[K^{1-\alpha }L^{\alpha }\right]$, where 1-α is the share of capital income and α is the share of labor income. Regarding the value of α, the existing literature usually sets it to 2/3 in empirical analysis, and Gollin (2002) studies have shown that the share of labor income almost does not change over time and space, which is about 2/3 [44]. In addition, many countries lack reliable data on the share of labor income. Based on this, according to the commonly used assumptions in previous literature, let α = 2/3. In terms of indicator selection, output Y is measured by real GDP in the Penn World Table (PWT) 10.0 version, capital stock K is measured by capital stock calculated by the perpetual inventory method in the Penn World Table, and labor input L is measured by the number of workers (working age between 15 and 64 years) in the World Bank WDI database. Among them, the actual GDP and capital stocks are calculated according to the constant price of 2011, the unit is millions of dollars. It should be noted that the estimated TFP also takes its natural logarithm into the model.

**Table 3 ijerph-19-03523-t003:** Linearity and remaining non-linearity tests.

Model		Model A	Model B	Model C	Model D
Threshold Variable		*lnPGDP*	*lnOPEN*	*lnIND*	*lnTFP*
Linearity Test	LM_F_ (H0: r = 0, H1: r = 1)	53.733 *** (0.000)	6.911 *** (0.000)	23.120 *** (0.000)	68.508 *** (0.000)
Remaining non-linearity Test	LM_F_ (H0: r = 1, H1: r = 2)	0.670 (0.512)	0.372 (0.690)	21.493 *** (0.000)	29.617 *** (0.000)
	LM_F_ (H0: r = 2, H1: r = 3)			−0.000 (1.000)	0.020 (0.888)
Model Tests	$F_{3}\left(H_{0}^{3}:B_{3}=0\right)$	9.710 *** (0.000)	3.204 ** (0.023)	14.075 *** (0.000)	38.935 *** (0.000)
	$F_{2}\left(H_{0}^{2}:B_{2}=0|B_{3}=0\right)$	29.166 *** (0.000)	3.420 ** (0.017)	6.196 *** (0.000)	23.657 *** (0.000)
	$F_{1}\left(H_{0}^{1}:B_{1}=0|B_{2}=B_{3}=0\right)$	6.063 *** (0.000)	0.248 (0.863)	2.391 * (0.067)	2.305 * (0.075)
Final Model Selection		m = 2, r = 1	m = 2, r = 1	m = 1, r = 2	m = 1, r = 2

Notes: ***, **, * denotes significance at the 1%, 5% and 10% level, respectively; *p*-values are in parentheses.

**Table 4 ijerph-19-03523-t004:** Linearity model and PSTR model estimation results.

Specification	Model A	Model B	Model C	Model D
Threshold Variable	*lnPGDP*	*lnOPEN*	*lnIND*	*lnTFP*
r	1	1	2	2
m	2	2	1	1
Parameter β_0_	0.252 *** (0.014)	0.269 *** (0.017)	0.741 *** (1.056)	−0.888 *** (0.110)
Parameter β_1_	−0.065 *** (0.007)	−0.038 *** (0.007)	0.000 *** (0.000)	−0.240 *** (0.020)
Parameter β_2_			−1.481 *** (2.112)	1.332 *** (0.122)
Location parameters $c_{1}$	6.558	4.1387	3.315	4.218
Location parameters $c_{2}$	9.675	4.1388	5.824	6.033
Slopes parameters $\gamma_{1}$	0.747	4.720	9.538	1.630
Slopes parameters $\gamma_{2}$			0.000	1.974
AIC criterion	−3.368	−3.301	−3.359	−3.452
Schwarz criterion	−3.342	−3.275	−3.322	−3.416
Number of obs.	944	944	944	944

Notes: *** indicate significance at the 1% levels; standard errors are in parentheses.

## Data Availability

The data used in this study are available on reasonable request from the corresponding author.

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
