# Peer review of "The Threshold Effect of FDI on CO2 Emission in Belt and Road Countries"

_ijerph, 2022, doi:10.3390/ijerph19063523_

Round 1
Reviewer 1 Report
Dear Authors!
Please include a separate literature review section in the study. Moreover, you should compare the results of this study with the previous studies.
Regards,
Author Response
Point 1: Please include a separate literature review section in the study. Moreover, you should compare the results of this study with the previous studies.
Response 1: Thank you very much for your advice. We have listed the literature review separately. In addition, the differences from the previous research are added in this paper, which indicates the research contribution of this paper. Specific contributions are as follows:
In this paper, we investigate the potential threshold effects on the relationship between FDI and CO2 emission. Thus, we propose to test the relevance to FDI regression parameters (or FDI-CO2 coefficients) into classes given the values of five main factors generally quoted in this literature: (i) per capital GDP, (ii) population density, (iii) degree of trade openness, (iv) industrialization and (v) total factor productivity. The approach has two main advantages. First, based on PSTR specifications, it derives FDI-CO2 coefficients, which vary not only between countries but also with time. Thus, it provides a simple parametric approach to capture both cross-country heterogeneity and time variability of the FDI-CO2 correlations. Second, the approach allows for smooth changes in country-specific correlations depending on a threshold variable. Nevertheless, no one has ever assessed the relative influence of each of these variables on FDI-CO2 correlations. On the contrary, our panel threshold regression framework allows establishing a “ranking” for the most frequently quoted explicative factors. Consequently, we consider the five potential threshold variables previously mentioned as potential explanations of the cross-country heterogeneity and/or the time variability of FDI-CO2 coefficients for the Belt and Road countries.
Reviewer 2 Report
The paper is based on FDI and carbon emission in Belt and Road Countries, however the methodology can be improved.
Through literature review, it can be identified that there is non linear relation between FDI and Carbon emission.
What is originality related to this front.
The author need to justify their methodology why such methodology has been used.
Literature review need to be addressed with newer studies.
Practical implication are required to be added in the paper.
Limitation section need to be intacted in current study.
Author Response
Response to Reviewer 2Comments
Point 1: The paper is based on FDI and carbon emission in Belt and Road Countries, however the methodology can be improved. Through literature review, it can be identified that there is non linear relation between FDI and Carbon emission. What is originality related to this front. The author need to justify their methodology why such methodology has been used.
Response 1: Thank you very much for your suggestion. We have improved the research method and proposed what is the main advantage of this method. As follows :
In this paper, we investigate the potential threshold effects of the relationship between FDI and CO2 emission. Thus, we propose to test the relevance of FDI regression parameters (or FDI-CO2 coefficients) into classes given the values of five main factors generally quoted from this literature: (i) per capital GDP, (ii) population density, (iii) degree of trade openness, (iv) industrialization and (v) total factor productivity. The approach has two main advantages. First, based on PSTR specifications, it derives FDI-CO2 coefficients, which vary not only between countries but also with time. Thus, it provides a simple parametric approach to capture both cross-country heterogeneity and time variability of the FDI-CO2 correlations. Second, the approach allows for smooth changes in country-specific correlations depending on a threshold variable. Nevertheless, no one has ever assessed the relative influence of each of these variables on FDI-CO2 correlations. On the contrary, our panel threshold regression framework allows establishing a “ranking” for the most frequently quoted explicative factors. Consequently, we consider the five potential threshold variables previously mentioned as potential explanations of the cross-country heterogeneity and/or the time variability of FDI-CO2 coefficients for the Belt and Road countries.
Point 2: Literature review need to be addressed with newer studies.
Response 2: Thank you very much for your suggestion. We have added the latest references, and due to the change of research methods, we have added more literature in the literature review section.
Point 3: Practical implication are required to be added in the paper.
Response 3: Thank you very much for your suggestion. In conclusion part, we have supplemented the practical implication of the study, as follows :
Our research found that, in general, attracting FDI has not been conducive to improving environmental quality in Belt and Road countries, and it has led to an increase in carbon dioxide emissions, in line with the pollution haven hypothesis. Belt and Road countries are mostly least developed or developing countries, and they are largely unable to attract foreign direct investment in high-tech industries. As the economic development levels of Belt and Road countries have risen, the promotional effect of FDI on carbon dioxide emissions has weakened. This may be because mature economies are better able to attract FDI and gradually attract more foreign investment in the field of renewable energy, thereby reducing the share of fossil fuels used in energy production and easing carbon dioxide emissions. Moreover, attracting FDI in the field of technology can lead to new technologies that completely change a country's energy sector and reduce carbon dioxide emissions. As a result, it is necessary to continue to promote economic development, actively attract foreign investment in high-tech industries, encourage foreign companies to invest in renewable infrastructure and modern technology, urge governments to provide subsidies to guide multinational companies toward green and clean technologies, and guide FDI toward high-tech industries and high-efficiency production. It is also necessary to require foreign investors to evaluate and publish information on carbon dioxide emissions during the investment process, to strictly control carbon dioxide emissions, and to promote environmental governance and sustainable development.
Point 4: Limitation section need to be intacted in current study.
Response 4: Thank you very much for your suggestion. The limitation of this study is that in the process of exploring the heterogeneity of FDI-CO2 coefficient, the overall scale of FDI is used, which is limited to data acquisition. This paper does not conduct in-depth research at the industry level. In fact, the impact of FDI in different industries on carbon emissions may be more realistic, and it will also make the mechanism of FDI-CO2 coefficient affected by a country ' s economic level, industrialization level, population density, opening level and technological level clearer. This is certainly our future research direction.
Reviewer 3 Report
This paper examines the nonlinear effects of FDI on countries situated along the Silk Road Economic Belt and the 21st century Maritime Silk Road on carbon dioxide emissions. The paper presents empirical findings and policy implications, however, the paper is not appropriate for publication due to following reasons.
First of all, the paper uses annual data of the variables from 2003-2014. It seems very outdated to finish the empirical study at the time point of eight years back in time from now. The sample period should better be extended.
Second, it is very unusual to see that all the variables (including lnGDP, lnCO2 and others) show stationarity as presented in table 3 in page 7. It should better be cross checked by using other panel unit root test. It seems that the data property was not investigated thoroughly before going to the empirical analysis. Besides, the authors continues to do the cointegration test after showing stationariy test results of the variables. It should be reminded that cointegration tests are done in case the variables are nonstionary, yet, to find a long-run relationship.
Third, the choice of the variable are not persuasive. Energy intensity is used as a proxy variable for technology level. It is not clear about how the energy intensity is defined and it is still not convincing to use this variable for technology level given that there exist other data such as total factor productivity which reflects technology level. Including trade openness and industry(?) variable (how this is defined is also unclear. I guess it would mean something about the industry structure) is not very convincing, either. Both of them should have a potential issue of multicollinearity. It should also be checked.
Fourth, one of the contribution of the paper seems to be about nonlinearity in the empirical study. However, it is questionable if this threshold approach using a statistical test is giving an advantage over doing something else. For example, categorising the economies in the study based on the income levels according to the World Bank definitions and see if the test results are different for those different income group countries. The linearity test used in the study does not specify more sophisticated groupings of the countries than the world bank definitions given that optimal number of transition function is 1 based on the linearity test.
Fifth, the interpretations of the empirical analysis results are very unclear. For example, take figure 3. The figure shows the carbon dioxide emissions, which is associated with the real GDP per person. Eventually, it is not FDI itself but the real GDP per capita is causing the carbon emissions. The figure is titled, yet, influence of coefficient of FDI on cabon dioxide emissions. The different channels that cause emissions have to be clearly presented by using correct and clear steps to unmask the channels. Morevoer, figure 5 clearly shows that the (average) elasticity of FDI to carbon dioxide emissions shows a downward trend. Would this mean that it is due to the economic development (that is measured by the real GDP per capita)? How can this be connected this to the threshold effects, then? By looking at the different economies in figure 6, it shows different patterns of these different economies (some show a decreasing trend, and some others show (almost) constant). How can these be explained?
Sixth, there are some minor points that need to be considered. The economies in the study include Afghanistan. This economy was in war during almost all the sample period. Was this considered? Is it appropriate to include this economy in the study? Or, was it not considered because it is one of the 62 economies in the study? The specification 4 has control variable X. It is a variable or variables? Specification (3) include real GDP per capita and population. Are these two all necessary? In page 4, there is a sentence saying “Most of the emerging economies located along[…]transitioning from heavy industry to light industry.” This sentence seems to be wrong.
Author Response
Response to Reviewer 3 Comments
Point 1: The paper uses annual data of the variables from 2003-2014. It seems very outdated to finish the empirical study at the time point of eight years back in time from now. The sample period should better be extended.
Response 1: Thank you very much for your suggestion. According to the World Bank WDI database, carbon emissions data were updated to 2018, so we updated the data to 2018, with a study interval of 2003-2018 and re-regression analysis.
Point 2: It is very unusual to see that all the variables (including lnGDP, lnCO2 and others) show stationarity as presented in table 3 in page 7. It should better be cross checked by using other panel unit root test. It seems that the data property was not investigated thoroughly before going to the empirical analysis. Besides, the authors continues to do the cointegration test after showing stationariy test results of the variables. It should be reminded that cointegration tests are done in case the variables are nonstionary, yet, to find a long-run relationship.
Response 2: Thank you very much for your suggestion. Based on all the questions and suggestions put forward by the reviewer, we have made great adjustments to the overall research framework of the paper. We consider five threshold models to estimate the FDI-CO2 coefficient, and refer to the study of Fouquau et al. ( 2008 ) without setting control variables. As per capital GDP, the level of trade openness, the level of industrialization, technical level, population density of five variables, respectively, into the PSTR model regression, forming five models, so delete the unit root and cointegration test. According to the suggestions of reviewer, we conduct descriptive statistical analysis on the variable characteristics selected in this paper.
Point 3: The choice of the variable are not persuasive. Energy intensity is used as a proxy variable for technology level. It is not clear about how the energy intensity is defined and it is still not convincing to use this variable for technology level given that there exist other data such as total factor productivity which reflects technology level. Including trade openness and industry(?) variable (how this is defined is also unclear. I guess it would mean something about the industry structure) is not very convincing, either. Both of them should have a potential issue of multicollinearity. It should also be checked.
Response 3: Thank you very much for your suggestion. First, we have used total factor productivity instead of energy intensity as the proxy variable of technical level, and supplement the calculation method in this paper. We have made a detailed description of the measurement methods of variables such as trade openness and industrialization level, as follows. In addition, as the paper changes the model, five variables are used as threshold variables, forming five models, so no longer need multiple collinearity test.
Carbon dioxide (lnCO2) emissions: the data for which is taken from the World Bank’s World Development Indicators (WDI) database. Carbon dioxide emissions are generated by the combustion of fossil fuels, including the consumption of solid, liquid, and gaseous fuels and combustion of natural gas, as well as during cement production. lnFDI: with data derived from the latest information released by UNCTADstat. The conversion variables (also the threshold variables) are: Level of economic development, which is represented by per capital GDP (lnPGDP) given in constant 2010 US dollars. PGDP data comes from the World Bank’s World Development Indicator (WDI) database. Population density(lnPOP) being calculated as the number of people per square kilometer of land area by dividing the size of the population by the land area; industrialization level is the value added by industry as a percentage of GDP; Openness (lnOPEN): exports plus imports divided by real gross domestic product per capital is the total trade as a percentage of GDP. Technology factor productivity(lnTEC):According to the C-D production function including capital stock K, labor input L and output Y, TFP is inversely solved, that is TFP = Y / [ K1-αLα ], where 1-α is the share of capital income and α is the share of labor income. Regarding the value of α, the existing literature usually sets it to 2/3 in empirical analysis, and Gollin (2002) studies have shown that the share of labor income almost does not change over time and space, about 2/3. In addition, many countries lack reliable data on the share of labour income. Based on this, according to the commonly used assumptions in previous literature, letα =2/3. In terms of indicator selection, output Y is measured by real GDP in the Penn World Table (PWT) 10.0 version, capital stock K is measured by capital stock calculated by the perpetual inventory method in the Penn World Table, and labor input L is measured by the number of workers (working age between 15 and 64 years) in the World Bank WDI database. Among them, the actual GDP and capital stock are calculated according to the constant price of 2011, the unit is millions of dollars. It should be noted that the estimated TFP takes natural logarithm into the model.
Point 4: One of the contribution of the paper seems to be about nonlinearity in the empirical study. However, it is questionable if this threshold approach using a statistical test is giving an advantage over doing something else. For example, categorising the economies in the study based on the income levels according to the World Bank definitions and see if the test results are different for those different income group countries. The linearity test used in the study does not specify more sophisticated groupings of the countries than the world bank definitions given that optimal number of transition function is 1 based on the linearity test.
Response 4: We have greatly adjusted the model and explained the advantages of the research method as follows :
In this paper, we investigate the potential threshold effects in the relationship between FDI and CO2 emission. Thus, we propose to test the relevance of FDI regression parameters (or FDI-CO2 coefficients) into classes given the values of five main factors generally quoted in this literature: (i) per capital GDP, (ii) population density, (iii) degree of trade openness, (iv) industrialization and (v) total factor productivity. The approach has two main advantages. First, based on PSTR specifications, it derives FDI-CO2 coefficients, which vary not only between countries but also with time. Thus, it provides a simple parametric approach to capture both cross-country heterogeneity and time variability of the FDI-CO2 correlations. Second, the approach allows for smooth changes in country-specific correlations depending on a threshold variable. Nevertheless, no one has ever assessed the relative influence of each of these variables on FDI-CO2 correlations. On the contrary, our panel threshold regression framework allows establishing a “ranking” for the most frequently quoted explicative factors. Consequently, we consider the five potential threshold variables previously mentioned as potential explanations of the cross-country heterogeneity and/or the time variability of FDI-CO2 coefficients for the Belt and Road countries.
Point 5: The interpretations of the empirical analysis results are very unclear. For example, take figure 3. The figure shows the carbon dioxide emissions, which is associated with the real GDP per person. Eventually, it is not FDI itself but the real GDP per capitall is causing the carbon emissions. The figure is titled, yet, influence of coefficient of FDI on cabon dioxide emissions. The different channels that cause emissions have to be clearly presented by using correct and clear steps to unmask the channels. Morevoer, figure 5 clearly shows that the (average) elasticity of FDI to carbon dioxide emissions shows a downward trend. Would this mean that it is due to the economic development (that is measured by the real GDP per capital)? How can this be connected this to the threshold effects, then? By looking at the different economies in figure 6, it shows different patterns of these different economies (some show a decreasing trend, and some others show (almost) constant). How can these be explained?
Response 5: As mentioned by the reviewer, economic development has a significant impact on carbon emissions, and some of this impact comes from FDI. When a country is in the underdeveloped stage of economic development, in order to attract FDI, it may reduce environmental regulation requirements, and attract investment projects are not conducive to carbon reduction. With the continuous maturity of economic development, the environmental requirements of FDI are increasing, and the carbon emissions of new FDI are reduced. In order to avoid the confusion of the influence mechanism, we re-examine the influence mechanism of economic development on FDI-CO2 coefficient, and put economic development as a threshold variable rather than an explanatory variable into the carbon emission equation to test the adjustment effect of economic development on FDI-CO2 coefficient. In addition, referring to the existing research, in addition to per capital GDP, our threshold variable also considers trade openness, industrialization level, population density and technological level, and examines the heterogeneous impact of a country’s multidimensional development characteristics on the FDI-CO2 coefficient. Finally, for the problem mentioned by the reviewers that the FDI-CO2 coefficients of different economies have obvious changes in different periods of time, considering the cross-country heterogeneity and time-varying nature of FDI-CO2 correlations, we use the above model to reanalyze the results as shown in figure 8 of the revised draft. The horizontal axis of this figure represents time, and the vertical axis is the FDI-CO2 coefficient. The reflected feature is that different countries attract FDI over time to influence carbon emissions. The coefficient heterogeneity is based on the different characteristics of FDI attracted by different economic stages, resulting in different carbon emissions. This is also the reason why this paper uses the PSTR model,based on PSTR specifications, we derive FDI-CO2 coefficients, which vary not only between countries but also with time. Thus, our work provides a simple parametric approach to capture both cross-country heterogeneity and time variability of the FDI-CO2 correlations.
Point 6: There are some minor points that need to be considered. The economies in the study include Afghanistan. This economy was in war during almost all the sample period. Was this considered? Is it appropriate to include this economy in the study? Or, was it not considered because it is one of the 62 economies in the study? The specification 4 has control variable X. It is a variable or variables? Specification (3) include real GDP per capital and population. Are these two all necessary? In page 4, there is a sentence saying “Most of the emerging economies located along[…]transitioning from heavy industry to light industry.” This sentence seems to be wrong.
Response 6: Thank you for your advice. First, the war issues mentioned by the reviewer are very important. We deleted Afghanistan and Palestine in the sample, and deleted East Timor considering the data acquisition, which is currently 59 sample countries. Since the revised paper takes GDP per capital and population density as threshold variables to build models separately, the multicollinearity problem is no longer involved here. Finally, the sentence errors mentioned by the reviewer have been revised.
Round 2
Reviewer 3 Report
Improvements are clearly observed in the revision draft compared with the original draft. However, to my view, there are still some remaining mistakes and unclarities which have to be either corrected or better specified.
First of all, there are mistakes in using terminologies/definitions in economics.
- There is no such thing as per capital GDP. It should be named as GDP per capita.
- The TFP from the C-D production function is wrongly written in page 7. It has to be corrected.
Secondly, be clear and consistent in using subscript notations. I see that j denotes different threshold points. Then, what is the subscript k in equation (4) in page 5? Moreover, there is no subscript i for beta in equations (1), (2), (3). Then, subscript i starts to show in equation (5). Go through the subscripts and see if they are all correctly written/used.
In page 8, two lines below the equation (9), it says that K is the number of explanatory variables. Where can we see different explanatory variables that are included in the empirical specification? It is very unclear to see how exactly the empirical model specification look, which was supposedly used for presenting results in a table that follows. For this reason, what does the table 3 show is not clear, either. For example, Model A in table 1, does it include all the other explanatory variables yet having the lnPGDP is included as a threshold variable? If so, it has to be clearly shown or connected to one of the specifications in page 8.
Figure 1 needs to be further clarified as well. Based on a common understanding of reading a graph on a plane, it is understood that FDI-C02 coefficients are dependent on these different macroeconomic variables. From which specification can it be shown that these bivariate relationships between FDI-C02 coefficients and each a macro indicator were modelled/investigated? It then goes back the previous comment about where are different explanatory variables in the specification. Or is it bivariate? To avoid these problems about unclarity, I think that the different stages of the PSTR should be better explained. Furthermore, unclarity regarding the figure 1 become even bigger with the title of the figure 1 which states 'influence coefficient of FDI on carbon dioxide emissions'. The title of the figure 1 is only confusing.
There are quite a few grammar errors and incomplete sentences. Language should still be improved. Some mathematical notations have mistakes, for example, as in equation 11.
